# Perceptions and Practice of Preconception Care by Healthcare Workers and High-Risk Women in South Africa: A Qualitative Study

**DOI:** 10.3390/healthcare9111552

**Published:** 2021-11-15

**Authors:** Winifred Chinyere Ukoha, Ntombifikile Gloria Mtshali

**Affiliations:** School of Nursing and Public Health, University of KwaZulu-Natal, Durban 4041, South Africa; mtshalin3@ukzn.ac.za

**Keywords:** preconception health, women’s health, healthcare workers, health promotion, healthcare provision

## Abstract

Preconception care is biomedical, behavioural, and social health interventions provided to women and couples before conception. This service is sometimes prioritised for women at high risk for adverse pregnancy outcomes. Evidence revealed that only very few women in Africa with severe chronic conditions receive or seek preconception care advice and assessment for future pregnancy. Thus, this study aimed to explore the perceptions and practice of preconception care by healthcare workers and high-risk women in Kwa-Zulu-Natal, South Africa. This exploratory, descriptive qualitative study utilised individual in-depth interviews to collect data from 24 women at high risk of adverse pregnancy outcomes and five healthcare workers. Thematic analysis was conducted using Nvivo version 12. Five main themes that emerged from the study include participants’ views, patients’ access to information, practices, and perceived benefits of preconception care. The healthcare workers were well acquainted with the preconception care concept, but the women had inconsistent acquaintance. Both groups acknowledge the role preconception care can play in the reduction of maternal and child mortality. A recommendation is made for the healthcare workers to use the ‘One key’ reproductive life plan question as an entry point for the provision of preconception care.

## 1. Introduction

Preconception care (PCC) entails the biomedical, behavioural, and social health interventions given to women and couples before conception, intended to improve their well-being by reducing behavioural and environmental factors contributing to poor maternal and child outcomes [1,2]. Most healthcare workers (HCWs) prioritize this service for women at high risk for adverse pregnancy outcomes [3]. However, evidence revealed that only very few women with severe chronic conditions that contribute to adverse pregnancy outcomes, such as rheumatic heart disease and diabetes, receive PCC advice and assessment for future pregnancy in Africa [4,5,6]. PCC recommendation in the maternity guidelines of South Africa proposes that all healthcare workers who care for women of childbearing age have the responsibility to encourage women to make reproductive choices and assist those considering pregnancy to improve their health and knowledge accordingly [7]. While PCC is care given to women before conception, antenatal care is routine care for women between conception and the onset of labour. In contrast, early antenatal care is antenatal care initiated within the first trimester [8].

The PCC experience and knowledge of healthcare workers and patients vary. A meta-synthesis of PCC experiences of women with diabetes shows some inconsistencies in PCC provision. Inconsistency regarding information given and a lack of integration between primary care where diabetes is managed and the specialist team providing PCC [9]. Women with epilepsy who are using sodium valproate experience many issues regarding PCC ranging from choosing not to have a baby due to failed medication switch to the risk valproate pose to the present pregnancy or chances of unsuccessful change from the medication before pregnancy. Therefore, a recommendation for a clear protocol for these women right from diagnosis was made [10]. A study revealed that improving the quality of the relationship between women with diabetes and the health care provider will enhance uptake of PCC among these women as they value the relationships and communication between them [11]. The PCC knowledge among Saudi Arabian women with diabetes was inadequate. There was a significant correlation between the level of education and PCC knowledge [12]. Health care providers are often confused on whose role it is to provide PCC services as general practitioners indicated that it is the responsibility of the nurses. In contrast, nurses assert that some aspect and complexity of PCC is beyond their scope [3,13]. Midwives agreed to the impact of PCC but acknowledged their role as the first point of contact for women who are already pregnant as too late for PCC intervention [13].

Many PCC interventions improved maternal and child outcomes, such as weight loss program, folic acid supplementation, multivitamin, and iron supplementation [14] PCC was proven effective in reducing the rates of congenital malformations and may also improve the risk of preterm delivery in women with type 1 and 2 diabetes [15]. It was also associated with improved pregnancy preparation and reduced risk of adverse pregnancy outcomes in women with pregestational diabetes [16]. A systematic review linking preconception and pregnancy care intervention effectiveness revealed that PCC in the form of general maternal health education effectively reduces neonatal and perinatal mortality. Although not significantly effective in maternal mortality reduction [17], PCC on contraceptive use and counselling and health optimization in some medical conditions will improve maternal mortality. Although a study linking the effects of PCC in improving pregnancy outcomes was inconclusive, it revealed a relationship between PCC education and counselling with improved knowledge and control of risky behaviours [18]. Several psychosocial risk factors such as teenage birth, history of mental health, behavioural conditions, and living in deprived areas were associated with low birth weight and other adverse infant outcomes [19], which can be prevented through PCC.

Several strategies have been identified that can be used to provide PCC effectively. It can be delivered in primary care and hospitals as inter-conception care, community-based outreach programs, and preconception care clinic settings [20]. Using current health care delivery systems to provide PCC in the form of nutrition and health behaviour improvement interventions could be more effective for an extended period. However, it requires social movement support [21]. PCC should be integrated to include men and women under one strategy [22]. Madanat and Sheshah [12] suggested that efforts be made to integrate PCC into several specialties in charge of women with diabetes, such as primary care physicians, family physicians, obstetricians, diabetologists, endocrinologists, and diabetes educators. An Iranian study of health care providers purports a lack of proper integration of PCC into other health care services as gaps in PCC service provision. It suggests the need for a PCC guideline and reminder for all health care workers attending to women of childbearing age [23]. The general practitioners and women suggested that practice nurses target PCC during routine contraceptive and cervical screening care [3,24]. Peer education strategy implemented in Dutch study to reach and educate women of childbearing age about PCC revealed that peer education is a feasible strategy for PCC implementation [25]. PCC should be integrated as a fixed protocol during maternity care, postnatal check-up, well-baby and immunization services, and contraceptive services [26]. Lassi, Dean [27] identified several ways of delivery of PCC. This includes within the education system as school reproductive health education programs, health care system through the primary level of care by community health workers, and other platforms such as community support groups, media campaigns, and workplace programs. Mitchell and Verbiest [28] describe novel strategies that will assist in reaching the population of women in the community with preconception health, such as using a computer-based PCC system and mobile technology through a public-private partnership.

Integrating PCC as a reproductive life plan (RLP) into contraceptive services was considered a feasible tool for promoting reproductive health, and thus its introduction was adopted by Sweden midwives [29]. The use of RLP to improve health outcomes among women and men of reproductive age has been identified as one strategy of delivering PCC [30]. Patients regarded RLP questions as crucial as some risk having an unplanned pregnancy due to lack of utilization of family planning. Therefore, the primary care provider should consider integrating reproductive life plan assessment into their services and connect patients to suitable contraceptives, preconception, and sexually transmitted infection services [31]. The effective delivery of PCC to women with diabetes is estimated to save the USA $5.5 billion in healthcare costs and lost productivity due to preterm delivery, congenital abnormalities, and perinatal mortality [32].

To increase PCC services provision and its integration into other programs for low and middle-income countries, a task-shifting to community health workers and partnering with communication and information technology to maximize the delivery and uptake of preconception interventions was suggested [33]. A study in South Africa exploring young women’s preferences of physical and mental health intervention strategies shows that they preferred community health workers as the delivery agent [34]. PCC provision is necessary for South Africa due to the persistently high level of non-communicable diseases [35]. Although PCC is a substantial aspect of women’s health, the delivery and utilization of PCC are highly problematic due to a lack of understanding of the concept among HCWs and women. The HCW’s and women’s perceptions of PCC can influence how they provide and utilise PCC. There is a paucity of evidence of PCC practices and perception in South Africa; therefore, this study will investigate this aspect from the perspective of healthcare workers and women.

## 2. Aim

The study aimed to explore the perception and practice of preconception care by healthcare workers and high-risk women in Kwa-Zulu-Natal, South Africa. In the context of this study, PCC practices entail the component of preconception care services delivered to the patients.

## 3. Research Methods

### 3.1. Study Setting

This study was conducted in a tertiary hospital with a dedicated preconception care clinic at eThekwini Metropolitan Municipality in KwaZulu-Natal province of South Africa. This tertiary hospital carters and gets referrals from the entire population of Kwazulu-Natal and parts of Eastern Cape province. eThekwini Metropolitan Municipality is the third most populated Municipality in South Africa after Gauteng and Cape Town, with 3.4 million inhabitants [36]. The Municipality has one tertiary and many district hospitals, including clinics. The province has the highest maternal and child mortality rate among women aged 10 to 24 years in South Africa [37]. This study is part of a mixed-method multiple case study conducted in a tertiary hospital and a higher education institution that trains nurses in the province. The study was conducted in the obstetric and gynaecological unit, which comprises preconception care, genetic, obstetrics, and gynaecology, feto-maternal medicine clinics. The selected tertiary hospital is a referral hospital, and the selected obstetric unit gets referrals of women with high-risk conditions.

### 3.2. Research Design

A qualitative descriptive design was used in this study to enable an in-depth [38] exploration of the perceptions of patients and healthcare workers (HCWs) regarding preconception care provision. This design was deemed vital in having a rich and diverse insight of the patient’s perspective and that of the health care workers to answer the questions of how and why regarding preconception care provision.

### 3.3. Research Participants

Data were purposively collected from two different groups of participants to obtain rich data. The participants included women at high risk of adverse pregnancy outcomes attending services in the obstetric and gynaecological unit and all health care workers attending to these patients. These health care workers were the providers of PCC in the unit and are a good representation of healthcare workers that high-risk patients would encounter for PCC because of their various clinical specialties. The participants were deemed appropriate for the study because they would be able to provide the necessary information regarding their PCC care provision.

Patients must have received at least one preconception care counselling or be at risk of adverse pregnancy outcomes, and health care workers must be employed permanently in the unit for at least one year to be eligible to participate in the study. Participants must be 18 years and older. The table below (Table 1) reveals the characteristics of the participants and the number of years they have been working or visiting the unit. Among the patients, ten are being investigated and managed for cardiac diseases/surgery, eight are being treated and investigated for infertility, and two each are being managed for hypertensive disorders, diabetes, and chromosomal abnormalities. The healthcare workers comprised of an obstetric and gynaecological and a feto-maternal medicine specialist. Two genetic nurses and a midwife specialist who is also a family planning nurse.

### 3.4. Sampling and Recruitment

A non-probability purposive sampling method was used to recruit a sample of patients and healthcare workers for this study. Twenty-four women at high risk of adverse pregnancy outcomes (*n* = 24) and specialist healthcare workers (*n* = 5) made up the sample size. The sample size was based on data redundancy. Potential participants were identified and approached through the assistance of the nurses in the unit. All women at high risk of adverse pregnancy outcomes that were attending various clinics in the obstetric and gynaecological unit during the data collection period and health care workers who attended to these women were approached for data collection. The managers of the hospital granted permission to access the unit and participant. The researchers visited the clinics on days when potential participants will be available to introduce the researcher, explain the study aims and objectives, and request their participation. Appointments for interviews were then made for data collection based on participants’ preferences.

### 3.5. Ethical Considerations

Ethical clearance for the study was obtained from the University of KwaZulu-Natal Human and Social Sciences Research Ethics Committee and the KwaZulu-Natal Health Research (HSSREC/00001069/2020) and Knowledge Management directorate reference number (KZ-202003-009). All the necessary institutional permission was obtained from the tertiary hospital prior to data collection. All protocols stipulated in the Helsinki declaration regarding voluntary participation, confidentiality, and anonymity were observed during data collection and reporting of findings.

### 3.6. Data Collection

The study data were collected using a semi-structured interview guide to conduct an in-depth interview between October and December 2020. There were open-ended qualitative items based on the participant’s perceptions and practice of PCC. Each group of participants was asked the same main questions with varying probing questions based on their responses. The main questions asked were as follows: “Please share with me your understanding of PCC? (What is this all about, what does it mean to you, and who and what does it involve).” “Tell me what you understand by reproductive life plan?” “Share with me your perceptions and views regarding PCC provision?” “Please share with me the aspect of PCC provided here.”

#### Interview Process

The researchers visited the unit twice before resuming the interview to get familiar with the environment and establish relationships with the participants. Interview appointments and participants preferred venues were established, and data were collected after obtaining voluntary informed consent. The interviews lasted between 25 to 60 min with few individual variations. Data were collected in English, with participants given the option to have the interviews in other preferred local languages. The individual in-depth interviews were deemed appropriate because they enabled the exploration of individual opinions that would have been hampered by focus group discussion. All interviews were audio-recorded, and field notes were made on each interview. Data collection and analysis co-occurred with transcription and were saved in a password-protected file using pseudonyms to maintain anonymity. Interviews were conducted by the first author and another doctorate student with experience in qualitative study after two days of training to acquaint her with the interview questions. Interviews were audio-recorded with permission from participants then transcribed verbatim while quotes were edited for clarity.

### 3.7. Data Analysis

Recorded interviews were transcribed verbatim. The authenticity of the transcripts was ensured by reading the transcripts while listening to the audio recording. Transcripts were sent to participants for correction to maintain the accuracy and credibility of the findings. Transcripts were then exported to Nvivo version 12 (QSR International, Doncaster, Australia) for organisation, coding, and analysis. The thematic analysis approach was used for data analysis following the six steps proposed by Braun and Clarke [39,40]. The six steps used include (a) reading through the data several times to enable immersion, (b) producing initial codes from the data, (c) sorting different codes to form themes, (d) reviewing and refining themes, (e) defining and naming themes (f) producing a report using data extracts in a concise, coherent, logical way. Data were coded by two independent coders and verified by the co-author. Modifications were made where necessary to provide clarity or to remove redundant words such as em and ehm. The Consolidated Criteria for Reporting Qualitative Research (COREQ) guideline was followed to report findings [41].

#### 3.7.1. Trustworthiness

The trustworthiness criteria proposed by Guba and Lincoln [42] were followed to ensure the quality of the study’s results. This includes credibility, dependability, confirmability, and transferability. All four aspects of trustworthiness were ensured as follows. Credibility relates to how the study analysis was conducted to ensure that no pertinent information was excluded [43]. Credibility was achieved through member checking with some research participants and with the co-investigators to confirm research findings and interpretation. Dependability, the extent to which data can be depended on over time, was ensured using an inquiry audit and using participant’s verbatim quotes to support findings interpretation. The use of two groups of participants and women with varying medical and surgical conditions helped the generalization of findings. At the same time, confirmability was ensured by involving two independent coders in the coding process to exclude subjectivity [44].

#### 3.7.2. Findings

Four main themes and 14 subthemes were generated concerning the provision of, and perceptions about PCC emerged from the data. They are views about PCC, services provided, patient’s access to PCC information, and perceived benefits of PCC and reproductive life plan (RLP). The summary of the emerged themes and subthemes is presented in Table 2.

##### Theme 1: Views about PCC

Four subthemes emerged with regards to participant’s perceptions about PCC. This includes preconception care as preventive care, which involves screening services, who PCC should be prioritised, and setting for PCC provision.

##### Subtheme 1.1: Preventive Care

Participants reported that PCC should be about providing services to women that will prevent complications that may contribute to adverse pregnancy outcomes and genetic disorders. PCC involves interventions to avert adverse pregnancy outcomes, especially for women with underlying conditions. It involves stabilising the conditions of the women before conception, advising, and educating them about their conditions so that they are aware of the implications of their medical and surgical conditions.

“*PCC focuses more on preventing genetic disorders and defects…it is about giving optimum care to a woman of childbearing age… a woman with medical conditions such as diabetes, cardiac, epileptic. What does it mean to her to have optimal control before she conceives? It is about giving advice, educating, and creating awareness.*”(G3)

“*Is the preventive care that we offer to women that are planning to fall pregnant… specifically here, we focus on women with high-risk conditions either from a previous history or family history or a condition that the patient themselves indeed have. We advise on whether it is safe or probably not safe to undertake a future pregnancy.*”(G1)

##### Subtheme 1.2: Screening for Risk Factors

It also includes investigative services to identify risk factors in a woman, family, or environment that may affect the pregnancy outcome. After this investigation, women are advised about the safety of future pregnancy and what they need to do if they are planning pregnancy.

“*This is a service provided to women before they fall pregnant so that they can have a planned pregnancy. Any questions about the pregnancy can be answered. If there are modifiable risk factors like drugs, alcohol, smoking, the weight, they can also be modified before they fall pregnant…*”(G2)

“*It is an opportunity to investigate them without the pregnancy because you can do much more without the concern about the pregnancy and then to advise them appropriately about the safety of falling pregnant both for the woman and for the fetus.*”(G5)

##### Subtheme 1.3: PCC Priority

All the HCWs reiterated that they prioritise women with chronic medical or surgical conditions for PCC. Although PCC is meant for everyone in the childbearing age, priority is given to those above 35 years and those at high risk of adverse pregnancy outcomes. This priority group is due to the shortage of human resources prevalent in the healthcare system and the number of women and men in the childbearing age bracket that should be seen.

“*Maternal age is considered—the very young and over 35. Women with prior medical conditions, like diabetes, hypertension, cardiac diseases, multiple miscarriages. People with epilepsy because we worry about control for them…**and*
*we worry about the medication they are taking, its effects on them and the outcomes on their babies.*”(G3)

“*…all women should have access to the PCC service, including women with medical and surgical disorders. Family history of certain genetic disorders…couples who have previous children with problems… those women attending contraception clinics and who have underlying problems. At the moment, we can’t offer it to all women because of the numbers…*”(G2)

##### Subtheme 1.4: PCC Provider

Participants stressed that PCC should be every healthcare worker’s business. All HCWs that attend to people of childbearing age should assume the PCC role. They should be able to ask them about their reproductive life plan then advise them accordingly. This is the same for HCWs that prescribe different teratogenic medications to people of the reproductive age. They should counsel them and ensure that they are using a reliable contraceptive method to avoid unplanned pregnancies that can be detrimental to both the mother and child.

“*It is every nurse’s job to be a health educator…it does not have to be only a designated genetic nurse or genetic counsellor.*”(G3)

“*…the family planning sister, youth, and adolescent Primary Health Care, and all HCWs should advise women about PCC.*”(G4)

“*Clinics are issuing Epilem to young patients, but no one is asking them which contraception they are on. If you take Epilem, you must be on very reliable contraception. You shouldn’t fall pregnant while on it, so all HCWs, including cardiology, rheumatology even dermatologists, should give PCC advice.*”(G2)

##### Subtheme 1.5: PCC Settings

The participants further stated that PCC should ideally start from home with good nutrition and be part of every primary health care clinic. It should also be provided in hospitals, family planning clinics, every specialist care that manages women at risk of adverse pregnancy outcomes. It should be included in the school curriculum.

“*PCC should start right from home…preconceptionally, people must be healthy and eat well.*”(G3)

“*PCC services should be done at the clinics because there are so many people. I have seen nurses in the waiting area talking to patients. Also, it should be provided at every regional hospital that has obstetrics and gynaecology service.*”(P11)

“*PCC service should actually start at the base hospital and the local clinics…the other place where this service should be part of family planning. I don’t think family planning is only about contraception; I think it should also have an element of women’s health. If you do stop your contraception, then you are planning to fall pregnant and should go and seek prenatal counselling.*”(G2)

“*It is quite okay for women who have had one pregnancy. Then maybe health care workers can counsel them about future pregnancies. However, the issue also comes about with first-time people who have not fallen pregnant before. Another issue will be the growing number of teenage pregnancies so it might be something that should be put into the school curriculum to inform them about these things.*”(G5)

##### Theme 2: PCC Practices

The HCWs stated that their PCC services include genetic counselling, PCC counselling and screening, preventive care, and contraceptive services.

##### Subtheme 2.1: Genetic Counselling Services

Participants confirmed that women with or suspected genetic issues are counselled in the unit for future pregnancies. In confirmed cases of a genetic disorder, this service is also provided for the children as premarital counselling and screening.

“*In genetics, the mothers are counselled about the birth defects that the babies can have…. We do test for the parents if there are chromosomal abnormalities or there are multiple miscarriages. I am opportune to see some of my clients as children and then as adults for genetic counselling.*”(G5)

“*Women who have had fetal anomalies, multiple miscarriages…I think miscarriages are under-recorded and under-scored in our health services. Any four five six first trimester miscarriages should be considered for PCC…you can then counsel them about prenatal supplements.*”(G3)

##### Subtheme 2.2: Screening Services

Participants also provided PCC services as investigations and screenings, which are carried out to determine future pregnancy risks, including folic acid supplementation.

“*I would advise them firstly checking to see if they have any medical issues so that we can get it under control. We do the regular screening. We do pap smears…we make sure they are at an ideal weight…If they are on any medication, they should continue taking it, but you need to check that they are safe before they conceive and also start folic acid at least three months prior to conception.*”(G2)

“*For women with a previous baby with abnormality, we test both parents for reoccurrence and advise.*”(G3)

##### Subtheme 2.3: Promotive, Preventive, Social Intervention, and Curative Services

They reported that they also attend and control any modifiable risk factors such as weight control and tobacco or alcohol/drug cessation. They also try to improve pregnancy outcomes by stabilising the patient’s condition and providing curative care before pregnancy. They also provide access to supplements and immunisations for women and educate them about the importance of those interventions.

“*During PCC, we would say don’t fall pregnant, or we would say fix this then fall pregnant or change this then fall pregnant or stop smoking and then fall pregnant…lose weight then fall pregnant……diabetic, control your sugars better and then fall pregnant the chances of having abnormalities are less.*”(G2)

“*PCC not only prevent women who are high-risk falling pregnant, but you also going to try and improve the outcomes of pregnancy by fixing things; for example, women with cardiac conditions should get surgery before they fall pregnant… a woman come to this clinic they can be examined.*”(G1)

“*We provide access to vitamins like folic acids, immunisations, alcohol, smoking, and recreational drugs…we educate them.*”(G3)

##### Subtheme 2.4: Contraceptive Services

The participants provide contraceptive services for pregnancy planning among women.

“*Here we also provide family planning services to women who are delivered here. We educate them about child spacing and to visit our pre-pregnancy clinic whenever they want to fall pregnant for further assessment. We encourage spacing of planned pregnancy.*”(G4)

##### Theme 3: Patients’ Access to PCC Information

Two themes emerged regarding women’s access to PCC information. This includes varying PCC awareness levels among women and poor reproductive life plan knowledge.

##### Subtheme 3.1: Varying PCC Awareness among Patients

There was varying awareness level among participants about PCC. Almost all the participants with cardiac conditions were aware of PCC. They were informed about what to do while planning pregnancy and immediately after conception, but that is not the case for other patients with other conditions, especially those treated for infertility. Most women have never heard about PCC, and even after the concept was explained to them, they confirmed that no one has ever informed them about it.

“*I knew about PCC; I was told about it when I had my first baby. I was given PCC counselling about my condition.*”(P15)

“*I haven’t heard about PCC anywhere before…. we just go to antenatal visits, and they check our blood pressure and others.*”(P12)

“*It is the first time I hear the word, so I am not sure what it means. It sounds like the care given to women.*”(P4)

##### Subtheme 3.2: Poor RLP Awareness

All the participants were not aware of the Reproductive Life Planning concept. A RLP tool involves one key question that HCWs should use to screen women’s reproductive plan and advise accordingly. However, that tool is not available or common among HCWs.

“*I have never heard about the reproductive life plan concept.*”(P2)

“*What does reproductive life plan mean? It is my first time to hear about it….*”(P6)

##### Theme 4: Perceived Benefits of PCC and Reproductive Life Planning (RLP)

The perceived advantage of PCC and RLP emerged as a theme with four subthemes of reduction in maternal and child mortality, prevention of birth defects, prevention of teenage pregnancy and pregnancy planning, and empowerment.

##### Subtheme 4.1: Reduction in Maternal and Child Mortality

The HCWs acknowledged the importance of PCC and RLP in reducing maternal and child morbidity and mortality.

“*PCC can save women’s life you may not see the benefits now. If you go through the system, you might find out that there are patients who may not want to fall pregnant because they realise that the risks are too high or maybe women where you can change the drug they are taking, and you prevent the abnormal babies, for example, women who are taking Epilem.*”(G2)

“*We are going to see better outcomes both mother and baby, and we probably will save a lot of mothers, or we would save mothers who are not supposed to be falling pregnant but are falling pregnant.*”(G3)

“*PCC will help us know what we are falling into and save our lives and the baby. If you know that you will be pregnant, you have to check your body every time to know what is going on in your body. Do you have BP and all the stuff…no woman will die. PCC service is good.*”(P23)

##### Subtheme 4.2: Prevention of Birth Defects

Most participants perceived PCC to assist in preventing congenital abnormalities through early screening, testing, and counselling.

“*PCC will assist in the prevention of fetal alcohol syndrome, which is the commonest quiet form of mental retardation…South Africa is number one in the world for fetal alcohol syndrome, and we will have a healthier, happier society.*”(G3)

“*Working in genetics, we can see the number of birth defects that could have been prevented if mothers were advised properly when they were planning their pregnancies…the preconception folic acid that can be given to prevent neural tube defects. All this will lead to better outcomes both for the mother and baby.*”(G5)

##### Subtheme 4.3: Pregnancy Planning and Reduction in Teenage Pregnancy

The participants perceived PCC and RLP as a valuable tool in preventing teenage pregnancy and in assisting with pregnancy planning. Although both the HCWs and women are not well acquainted with the concept of RLP, with a short explanation of the concept, they could perceive it as the ideal thing for the patients and HCWs.

“*PCC and reproductive planning are important because you can’t do anything without planning. Why do we fall pregnant without planning to? I think if you pre-plan your pregnancy, it will go well. With PCC, women are gaining information which is better because most of the time you get pregnant, which you don’t understand.*”(P3)

“*It is always best to know information before you go through anything. If the pregnancy is planned, you can never go wrong because you will have understanding.*”(P20)

“*PCC is very important, and there will be a reduction in child pregnancies and mishaps like disabilities…If they are educated my opinion is that it will be way better for them, for an older person is fine but for a young child pregnancy will be a burden… is so sad to see a teenager pregnant, if no one is there teaching them they just gonna be in trouble and continue falling pregnant.*”(P13)

“*RLP sounds nice…the ideal would be RLP because, unfortunately, the challenges that we are still facing in KwaZulu-Natal and many parts of the country is that patients don’t have a plan.*”(G3)

##### Subtheme 4.4: Empowerment

PCC was perceived as essential in equipping women with knowledge and empowering them to handle their conditions. This empowerment can be in inform of educational resources and knowledge that will enable them to take charge of their life and reproduction.

“*PCC rewards to the patients are more empowered to control their condition and their lives.*”(G2)

“*It will make me act better as I plan to begin my family. it will give women power because even if you are pregnant, you can be able to save your baby, understand better and feel equipped.*”(P3)

“*…they are more aware of their condition, and patients will be more prepared for the future.*”(G3)

## 4. Discussion

The findings of this study reveal that most HCWs felt that integral to PCC is preventive and screening services which are rendered to women at high risk of adverse pregnancy outcomes. These two subthemes gave a clue that HCWs understand the overarching aim of PCC. Van der et al. [45] argued that PCC is deeply rooted in the ethical principle of nonmaleficence. Therefore, the prevention from the harm of future children is the primary goal of PCC and collective responsibility of both prospective parents. However, PCC should be for all women and men of childbearing age. The HCWs further revealed that currently, they could only provide these services to women with underlying health problems. This is primarily due to the shortage of human resources inherent in the healthcare system. General practitioners in the UK focused preconception advice on women with existing medical conditions and prefer to discuss contraception with women of childbearing age instead of preconception health [3]. HCWs describe PCC’s provision as every healthcare worker’s business. They reveal that PCC should be part of every primary health care clinic, family planning clinic, hospital, and part of the high school curriculum. Thus, PCC should be a joint venture involving all stakeholders across all settings, from the home to tertiary settings. Every healthcare worker who attends to women at their childbearing age should enquire about their reproductive plan and counsel appropriately. PCC has been described as a collaboration involving gynaecologists, midwives, general practitioners, dieticians, pharmacists, physiotherapists, nurse practitioners, other specialists, preventive child health care, and social workers [26]. According to Dean et al., all HCWs can and should provide PCC to women and couples [46]. Hurst and Linton [47] argued that the provision of PCC to women and men by identifying risk factors and encouraging healthy living is the vital role the primary health care workers.

The study revealed inequality in access to PCC information among patients at high risk of adverse pregnancy outcomes. It shows that women with diagnosed cardiac conditions had better access to information about preconception care services than others. This may be because this group is further prioritised for PCC due to the teratogenic effects of most of these women’s medications. According to the saving mothers report in South Africa, Hypertension is the leading direct cause of maternal death followed by non-pregnancy-related diseases such as cardiac conditions [48]. This lack of PCC awareness was more prominent among women being treated in the unit for infertility. Notwithstanding that infertility offers a window of opportunity to provide PCC, participants being investigated for infertility in the study were not offered PCC. This is consistent with an Iranian study that found that women treated for infertility were not offered all the necessary PCC intervention [49]. PCC and lifestyle modifications are essential factors in in vitro fertilization outcomes [50], and preconception risk assessment of infertile couples can enhance the chance of treatment outcome [51]. Hence PCC services can be successfully integrated into fertility treatment programs. The authors of [52] argued that preconception lifestyle advice should be provided to people presenting with infertility and proposed for a guideline to be developed to enable clinicians to implement PCC during infertility treatment. These patients can be encouraged to undertake these services before fertilization treatment [53]. All the participants were unaware of the reproductive life plan concept. Few HCWs stated that they ask women about their reproductive life plan. However, they do not routinely use the RLP question to counsel women of childbearing age. RLP is not a concept that HCWs and women of childbearing age in many developing countries are familiar with. The use of One Key Question, which the Oregon Foundation for Reproductive Health developed, is suggested to improve PCC provision. It provides an entry point for the health providers to routinely screen for pregnancy intentions among women by asking, “Would you like to become pregnant in the next year.” This question would enable the health provider to decide whether to provide PCC or other preventive reproductive health services such as contraceptive services for the women [54]. Furthermore, women’s reproductive plan is necessary to deliver chronic disease care as PCC implementation and should not necessarily be viewed as a new addition to given care. Instead, it should be synonymous with providing evidence-based care to reproductive-age people [55]. Providing PCC to women treated for infertility in this unit will make a considerable difference as their PCC awareness and utilization were inadequate.

Strategies used for PCC delivery by HCWs include genetic counselling, screening services, PCC counselling, preventive services, and family planning. These are some but not all the PCC components. The HCWs were very knowledgeable about various components and PCC packages. PCC interventions by the WHO include but are not limited to genetic conditions, nutritional assessment, tobacco use, environmental health, infertility, unwanted pregnancy, vaccine, and many others [56]. The HCWs in the current study practiced most of the components of PCC. They, however, omitted the aspect of mental health, infertility, and violence. A previous study in South Africa revealed that young people preferred integrating mental health as a significant part of preconception strategies instead of targeting only their physical health [34].

Participants believed that if provided adequately, PCC could reduce maternal and child mortality and prevent congenital abnormalities. PCC was advocated for Africa as a significant intervention to reduce maternal mortality rates. In contrast, many high-income countries introduced it to encourage pregnancy planning [57,58]. They also envisage that it will lead to pregnancy planning and the prevention of teenage pregnancy. WHO advises that countries should invest in PCC to prevent unintended pregnancy among all ages [56]. Both HCWs and patients perceive that PCC will lead to women’s empowerment. PCC provision has been described as improving patients’ knowledge by empowering them to seek advice and modify any risk factor for improved health before pregnancy through mass media and campaigns promoting PCC [3]. Improving the knowledge of women with chronic conditions through PCC provision was purported to empower them about the preventable risks involved with conception with uncontrolled conditions and make them aware of pregnancy planning choices available to them [59,60]. The current study was conducted in a referral hospital for high-risk patients; therefore, the patients’ and healthcare workers’ perceptions of PCC in this area of care might differ from those in the lower levels of care. Health care providers’ and prospective mothers’ perceptions about PCC influences its provision and utilization [61]; therefore, the perception and practice of PCC among participants in this study may be different from those in other settings as these are high-risk patients.

## 5. Conclusions

The findings of our study revealed that the HCWs were well oriented to PCC and appreciate its role in the reduction of maternal and child mortality and morbidity. However, this is not the case with most women at high risk of adverse pregnancy outcomes. It is noteworthy that the women, in this case, are at high risk of adverse pregnancy outcomes while the HCWs were from a tertiary institution and are responsible for PCC provision. The PCC practice of HCWs was consistent with their awareness level as they deliver most PCC components based on their area of specialty. Nevertheless, research has shown that infertility treatment provides a window of opportunity for PCC provision. This is because lifestyle modification and PCC were recognised as essential factors in infertility treatment outcomes yet women treated for infertility were not offered PCC. Both the women and HCWs have an excellent perception of the role of PCC and reproductive life planning in improving pregnancy outcomes. However, there is poor RLP awareness among both groups, and very few HCWs used the RLP key question to assess the PCC need for clients.

### 5.1. Limitations of the Study

The main limitation of this study is that although the World Health Organisation proposed that PCC should target both men and women as men’s health and behaviour have implications for their partner and their unborn children. Men have a vital role as husbands, fathers, and community members [62,63]. This study was conducted among women only. Another limitation is the small group of participants involved in the study. The findings from this selected group should be used with caution when compared with the general population of women and healthcare workers. There was also no advanced statistical analysis involved in this study qualitative study.

### 5.2. Recommendations

Further study should include men to investigate what type of PCC care is provided to them and their perceptions. In addition, the RLP concept should be introduced to women and HCWs, and the effectiveness of RPL questions in investigating the reproductive plan of women should be accessed. We also recommend that women being treated for infertility should not be overlooked during the PCC provision.

## Figures and Tables

**Table 1 healthcare-09-01552-t001:** Participants’ characteristics.

Pseudonym	Position in the Unit	Number of Years in the Unit
Healthcare Workers		
G1	Specialist medical officer	15 Years
G2	Specialist medical officer	13 Years
G3	Specialist Nurse	43 Years
G4	Specialist Nurse	14 Years
G5	Specialist Nurse	8 Years
**Patients**	**Reasons for consultation**	**Number of Years visiting the Unit**
P1	Cardiac Patient	7 Years
P2	Cardiac Patient	4 Years
P3	Patient with infertility	2 Years
P4	Patient with infertility	3 Years
P5	Patient with infertility	2 Years
P6	Patient with infertility	10 Years
P7	Cardiac Patient	2 Years
P8	Cardiac Patient	7 Months
P9	Patient with infertility	1 Year
P10	Patient with infertility	7 Years
P11	Patient with infertility	5 Years
P12	Patient with infertility	5 Years
P13	Patient with a chromosomal abnormality	2 Years
P14	Patient with a chromosomal abnormality	1 Year
P15	Cardiac patient	6 Years
P16	Cardiac patient	12 Years
P17	Hypertensive patient	1 Year
P18	Cardiac patient	1 Year
P19	Diabetic patient	5 Months
P20	Cardiac patient	4 Years
P21	Diabetic patient	1 Year
P22	Cardiac patient	1 Year
P23	Cardiac patient	1 Year
P24	Hypertensive patient	1 Year

**Table 2 healthcare-09-01552-t002:** Summary of the emerged themes and sub-themes.

Themes	Sub-Themes
Views about PCC	Preventive care
	Screening for risk factors
	PCC priority
	PCC settings
PCC services provided	Genetic counselling services
	Screening services
	Promotive, preventive, social interventions, and curative services
	Contraceptive services
Patients access to PCC information	Varying PCC awareness level among patients
	Poor reproductive life plan awareness
Perceived benefits of PCC and RLP	Reduction in maternal and child mortality
	Prevention of birth defects
	Pregnancy planning and reduction in teenage pregnancy
	Empowerment

PCC: preconception care; RLP: reproductive life plan.

## Data Availability

Data from this qualitative study is the property of the University of KwaZulu-Natal and may be made available upon request from the University or the study authors.

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
