# Peer review of "Perceptions and Practice of Preconception Care by Healthcare Workers and High-Risk Women in South Africa: A Qualitative Study"

_healthcare, 2021, doi:10.3390/healthcare9111552_

Round 1
Reviewer 1 Report
I am happy with many of the edits the authors have done here.
My main comments are:
- Defining pre-conception care:
Preconception care is identifying modifiable risk factors before pregnancy (and conception) and provide biomedical, behavioural, and social health intervention that improve maternal and health outcomes.
Interventions after conception are part of early pregnancy care and strictly speaking not pre-conception.
Many interventions that begin pre-pregnancy can continue during pregnancy.
This needs to be clear, better defined in the paper and not confused.
- The discussion surrounding paternal preconception care is based on a paper that primarily discussed smoking cessation but does not provide good evidence that it is important – the paper discusses ethics but does not provide outcome data. While smoking cessation might improve pregnancy rates, the authors should provide better clinical evidence here.
- The systematic review and meta-analysis the authors referenced did not find an improvement in maternal mortality with PPC. However, one can argue that in some medical conditions such as cystic fibrosis and cardiac disease, preconception care that also includes contraceptive use and counselling as well as optimising their medical condition will improve maternal mortality.
- The study investigated the awareness of a very select group of patients and staff in a tertiary unit who are far more likely to have encountered PPC. As such, they have not achieved their aim to ‘explore the perception and practice of preconception care by healthcare workers and women in Kwa-Zulu-Natal, South Africa’. The authors should consider rephrasing their aim to reflect the specific demographic of their participants. They should also acknowledge this in their conclusion.
- The authors argue early on that all women should receive PPC. However, their study includes only high-risk women. Over emphasising gender bias and inclusion of low-risk women in the introduction gives the impression that they are planning on investigating these aspects in their study. Perhaps leading with the provision of PPC in high-risk women and describing its benefits would be better.
- The paper needs to be read through again. Some areas where there have been edits have grammatical errors and typos.
Reviewer 2 Report
the manuscript has been redrafted and meets the criteria for publication in its current form
Author Response
Thank you
Reviewer 3 Report
I consider that authors adequately response to my observations
Author Response
Thank you for reviewing this manuscript.
Reviewer 4 Report
Dear Authors,
You have done a great job. I have no other comments.
Author Response
Thank you so much for reviewing our manuscript.
This manuscript is a resubmission of an earlier submission. The following is a list of the peer review reports and author responses from that submission.
Round 1
Reviewer 1 Report
Overall
This is an interesting narrative of a selective group of patient’s and healthcare worker’s knowledge about PPC.
Introduction
Is there a national standard for the provision of PPC in South Africa?
The authors state that PPC should be provided to all pregnancies during or before pregnancy irrespective of risk. However, the referenced paper by Dean discusses a framework of PPC to identify modifiable and non-modifiable risk factors that can be addressed before or during pregnancy to improve outcomes – this should be mentioned in the introduction. They also show a reduction in neonatal morbidity and improved breastfeeding as measures in the metanalysis. Interventions because of PPC included smoking cessation and taking folic acid. The authors then state that selective provision for chronic health conditions is not helpful in Africa but the references used are qualitative works that discuss knowledge and attitudes, awareness, and practises but not clear data demonstrating ineffective PPC in those groups. Their statement is contrary to published work and I recommend the authors rephrase the opening paragraph to reflect the above comments.
Systematic review and meta-analysis of the effectiveness of pre-pregnancy care for women with diabetes for improving maternal and perinatal outcomes. Wahabi HA, Fayed A, Esmaeil S, Elmorshedy H, Titi MA, et al. (2020) Systematic review and meta-analysis of the effectiveness of pre-pregnancy care for women with diabetes for improving maternal and perinatal outcomes. PLOS ONE 15(8): e0237571. https://doi.org/10.1371/journal.pone.0237571
Effectiveness of a Regional Prepregnancy Care Program in Women With Type 1 and Type 2 Diabetes. Helen R. Murphy, Jonathan M Roland, Timothy C. Skinner, David Simmons, Eleanor Gurnell, Nicholas J. Morrish, Shiu-Ching Soo, Suzannah Kelly, Boon Lim, Joanne Randall, Sarah Thompsett, Rosemary C. Temple
Diabetes Care Dec 2010, 33 (12) 2514-2520; DOI: 10.2337/dc10-1113
Associations between pre-pregnancy psychosocial risk factors and infant outcomes: a population-based cohort study in England. Harron, Katie et al. The Lancet Public Health, Volume 6, Issue 2, e97 - e105
The authors may also consider distinguishing between early antenatal care and pre-conception care as concepts. How about linking pre-pregnancy and pregnancy care?
Effectiveness of Continuum of Care—Linking Pre-Pregnancy Care and Pregnancy Care to Improve Neonatal and Perinatal Mortality: A Systematic Review and Meta-Analysis
Kikuchi K, Okawa S, Zamawe COF, Shibanuma A, Nanishi K, et al. (2016) Effectiveness of Continuum of Care—Linking Pre-Pregnancy Care and Pregnancy Care to Improve Neonatal and Perinatal Mortality: A Systematic Review and Meta-Analysis. PLOS ONE 11(10): e0164965. https://doi.org/10.1371/journal.pone.0164965
I recommend the introduction include a paragraph with references showing where strategies delivering PPC have a proven improvement in perinatal outcomes for mothers and their babies. What does the literature say is the best possible provision of PPC?
The next 2 introduction paragraphs discuss patient and healthcare workers experiences and knowledge about PPC, which is appropriate for the manuscript aims but the authors should state that this is what they are introducing – perhaps with an opening statement to the section.
The debate surrounding who and how PPC should be delivered is again important. Is there any evidence that compares different settings for delivery of PPC and their effect on perinatal outcomes?
Family planning is usually considered part of PPC in healthcare since this enables patients to optimise their healthcare through interventions before embarking on a pregnancy.
Methods
Qualitative interviews were a reasonable method. Were all participants asked the same questions? Who performed the interviews?
The authors mention that ‘These patients and health care workers were deemed appropriate for the study with the rationale that they have or should have encountered PCC’. However, is it appropriate to make this assumption? For example, is standard practise in South Africa that the healthcare workers deliver PPC or that the patients are offered PPC?
I understand that the authors have stated that all patients should receive preconception counselling but as mentioned above the intention is to identify risk factors for interventions and health improvement to reduce adverse perinatal complications. It seems unlikely that patients being seen with infertility are the most suitable patients to enquire about the provision of PPC. 33% of their patients were infertility patients sampled from a gynaecology clinic. Can the authors explain their inclusion? Most healthcare improvements in this group (e.g., stop smoking and reduce alcohol) is intended to improve fertility (although they are also recognised PPC interventions although not the intention in this setting). Equally, only 5 healthcare workers were included but more detail regarding their clinical roles is required to justify their inclusion – would they be expected to provide PPC? Are they a good representation of healthcare workers patients would encounter for PPC? Why?
The remainder of the methods were described well.
Results
Results section is well written.
Discussion
The discussion is fine and describes the differences between patient groups and their exposure to PPC. However, if the standard of care model in South Africa favours PPC for chronic medical conditions, the lack of PPC for fertility patients is not surprising.
Is PPC described by the authors, in fact the provision of general health/lifestyle interventions identified as important for all women of childbearing age? Should this be a national target? Does the literature support this notion?
Reviewer 2 Report
The article describes a very important medical problem in African countries regarding preconception care. Of course, this care should be provided by qualified medical personnel, doctors and midwives. The authors emphasize that the study was conducted in an area CHARACTERIZED BY A VERY HIGH MORTALITY AMONG BORNING WOMEN AND CHILDREN BETWEEN 10 AND 24 YEARS OF AGE. Since this is such an important issue in this community, I have to ask 3 key questions: 1 why only adult women were included in the study (apart from its significant methodological imperfections) 2.why only 5 people of medical staff took part in the study (as I understood only 2 doctors ( 3. why only 24 women participated in the study These three questions strike out any value of this work, no further comments are necessary. This manuscript is not suitable for publication, even taking into account a specific ethnic group.
Reviewer 3 Report
The present manuscript aims to explore the perception and practice of preconception care by health workers and women in Kwa-Zulu-Natal, South Africa. I consider it to be a well-designed study and, in general, adequately written, some observations that I consider important are 1.- I consider that both in the title and the conclusion it should be noted that they are high-risk women 2.- The discussion should address the fact that it is a referral hospital for high-risk patients, which may be very different from the perception in first-level units 3.- It is not compared with other similar studies in the discussionAuthor Response
Please see the attachment

Reviewer 4 Report
Dear Authors,
The presented study tackles an important issue of Perception and Practice of Preconception Care by Healthcare Workers and Patients in South Africa. The study was conducted in an environment which is specific in terms of the culture, which constitutes an additional value of the paper. The study was conducted reliably with the appropriate selection of statistical tests.
However, some issues require complementary information:
- I suggest incuding in Limitation of the study the information about small group and that there was no advanced statistical analysis.
- I suggest dividing Table 1 on two tables – one with position in the unit of the healthcare workers and the second one with the reason of consultation of the patients . Moreover I suggest incuding in the text the information about the number of participants in accordance with the reason of consultation.
